# Validity of the Baiobit Inertial Measurements Unit for the Assessment of Vertical Double- and Single-Leg Countermovement Jumps in Athletes

**DOI:** 10.3390/ijerph192214720

**Published:** 2022-11-09

**Authors:** Federica Camuncoli, Luca Barni, Sebastiano Nutarelli, Jacopo Emanuele Rocchi, Matteo Barcillesi, Irene Di Dio, Andrea Sambruni, Manuela Galli

**Affiliations:** 1Department of Electronics Information Technology and Bioengineering, Politecnico di Milano, 20133 Milan, Italy; 2E4Sport Lab, Politecnico di Milano, 23900 Lecco, Italy; 3Department of Physiotherapy, Faculty of Health Sciences, University of Malaga, 29071 Malaga, Spain; 4Service of Orthopaedics and Traumatology, Department of Surgery, (Ente Ospedaliero Cantonale) EOC, 6900 Lugano, Switzerland; 5School of Public Health, Physiotherapy and Sports Science, University College Dublin, D04 V1W8 Dublin, Ireland; 6Department of Movement, Human and Health Sciences, University of Rome Foro Italico, 00135 Rome, Italy; 7Villa Stuart Sport Clinic—FIFA Medical Centre of Excellence, 00136 Rome, Italy; 8Department of Medicine, Surgery and Neuroscience, Università degli Studi di Siena, 53100 Siena, Italy

**Keywords:** countermovement jump, vertical jump, inertial sensors, single-leg jump, Bland–Altman, Baiobit

## Abstract

Jump tests are simple, quick to execute, and considered the most reliable tool to measure lower extremities power and explosiveness in athletes. Wearable inertial sensors allow the assessment of jumping performance on any surface. The validity of inertial sensors measurements is a pivotal prerequisite to reliably implement their utilization in the clinical practice. Twenty-seven athletes (20 M/7 F, age: 27 ± 7 years old) performed five double-leg countermovement jumps (CMJs) and three single-leg CMJs per side with their hands on their hips. Jump height was measured/computed simultaneously with the optoelectronic system, force platforms, and the Baiobit inertial sensor system. The athletes completed the international physical activity questionnaire (IPAQ). When comparing the methods (Baiobit vs. force platforms), a non-statistically significant bias of 1.8 cm was found for two-leg CMJs and −0.6 cm for single-leg CMJs. The intraclass correlation coefficients (ICCs) was “excellent” for double-leg CMJs (ICC = 0.92, 95% CI = 0.89–0.94) and “good” for single-leg CMJs (ICC = 0.89, 95% CI = 0.85–0.91). When comparing the methods (Baiobit vs. force platforms + optoelectronic system), a non-statistically significant bias of −0.9 cm was found for two-leg CMJs and −1.2 cm for single-leg CMJs. The intraclass correlation coefficient (ICC) was “good” for both double-leg CMJs (ICC = 0.80, 95% CI = 0.73–0.85) and for single-leg CMJs (ICC = 0.86, 95% CI = 0.80–0.89). Baiobit tends to overestimate double- and single-leg CMJ height measurements; however, it can be recommended in the world of rehabilitation and sport analysis.

## 1. Introduction

Jump tests are simple, quick to execute, and considered the most reliable tool to measure power and explosiveness in athletes at lower extremities [1]; in fact, both in individual and team sports, vertical jump performance represents an important prerequisite for optimal performance during competitions and a fast option evaluate the knee status at the time to return to sport after anterior cruciate ligament reconstruction [2,3]. This is particularly relevant for single-leg jumping tests since a variety of sports injuries occur during single-leg motor tasks, including breaking then changing direction, landing, and pivoting [4]. Physical therapists’ intra-rater reliability in the qualitative visual assessment of lower limb movements during the execution of functional tests ranged from slight to almost perfect and the inter-rater reliability ranged from fair to good [5]. Noticeably, these agreements tended to increase with a higher clinical experience level combined with the implementation of dichotomous ratings [5]. However, this does not consider the quantitative evaluation [5] that musculoskeletal screening should include. This further step should be supported by dedicated devices to quantify parameters such as the range of motion, muscle strength and related potential imbalances, and various performance indexes (e.g., a jump height or a hop distance in cm) to reliably inform clinicians and eventually encourage them to design effective interventions which could possibly decrease injury risk [6].

Regarding this aspect, the clinical applications of monitoring human biomechanics benefit from the integration of engineering knowledge and the continuous development of new technologies [7]. An assessment via biomechanical and stereophotogrammetry-based human movement analysis systems is useful to address clinical issues in different patient subgroups, such as orthopedic and athletes; for this reason, musculoskeletal and sports physical therapists should familiarize themselves with the everyday implementation of such devices [7] and gain the required skills to use reliable and validated devices and protocols in their everyday clinical practice.

In clinical practice, it must be considered that some tools which are commonly used to measure motion and applied forces are not recommended to evaluate these aspects unless validated and accompanied by operational protocols, thus ensuring the quality and accuracy of the desired measurements [8,9].

Although jump tests can be conducted using contact mats, infrared platforms, and accelerometers, force plates reliably record quantitative parameters such as ground reaction forces, flight times, and jumped heights, providing the most accurate and reliable measurements [1].

In combination with this, optoelectronic systems represent the technological gold standard used in motion analysis for a kinematic evaluation [10], but these final systems are unfortunately expensive and not easily transportable; therefore, they are not useful to clinicians aiming to reliably assess human movement.

On the contrary, wearable inertial sensors are cheaper and transportable; hence, they are available to administer, both in clinics and for pitch-side tests. Moreover, this technology allows the assessment of jumping performance on any surface (e.g., firm ground, sand, or grass).

This stated, the validity of inertial sensors measurements is a pivotal prerequisite to reliably implementing their utilization in clinical practice.

Vertical jump quantitative motion measurements with inertial sensors have been validated already [3,11,12,13,14] and a further study investigated the concurrent validity of inertial sensors compared to force plates for an assessment of the flight times of counter movement jumps (CMJs) in professional male soccer players (18 ± 3 years), showing excellent intraclass correlation coefficient (ICC) values (0.92–0.97) [15].

In this study, among all sensors available on the market, the Baiobit inertial sensor system (Rivelo Srl—BTS bioengineering Group, Milan, Italy) was used.

Due to differences in hardware and/or software, the validity of an inertial sensor system is not transferable to another peer device, even though it uses the same technology.

Thus, the present study aimed to test the simultaneous validity of the utilized wearable inertial sensor system for two-leg and single-leg CMJ height assessment versus the gold standard method for motion analysis using the optoelectronic system [10] in order to possibly empower clinicians with a new portable, wearable, and cheaper tool to assess, track, and monitor human movement, thus verifying our hypothesis that the inertial sensor used can be a valid and useful tool in the world of rehabilitation and sports performance.

## 2. Materials and Methods

### 2.1. Subjects

In total, 27 athletes were recruited for the tests (20 males and 7 females, age: 27 ± 7 years, height: 1.86 ± 0.09 m, weight: 75 ± 12 kg, body mass index (BMI): 21.7 ± 2.0 kg/m^2^) (Table 1). Participants were included in the study if they: (i) trained for a minimum of 4 h per week, (ii) were aged between 18 and 40 years, (iii) had a BMI between 18.5 and 24.9 kg/m^2^, and (iv) had no self-reported neurological or musculoskeletal disorder.

An informed consent was obtained for all the study participants and the Ethical Committee of Politecnico di Milano approved the research (submission no. 22/2021) and all tests were carried out at the “Luigi Divieti Posture and Movement analysis laboratory”.

### 2.2. Devices

The instrumentation used was the following: A Baiobit sensor, comprising a triaxial accelerometer with multiple sensitivity levels (±2, ±4, ±8, and ±16 g); a 13-bit triaxial magnetometer (±1200 µT); and a triaxial gyroscope with multiple sensitivity levels (±250, ±500, ±1000, and ±2000°/s), manufactured by Rivelo Srl—BTS bioengineering Group, Milan, Italy. The Baiobit sensor works with an accelerometer frequency bandwidth ranging from 4 to 1000 Hz, a gyroscope bandwidth ranging from 4 to 8000 Hz, a magnetometer bandwidth up to 100 Hz, and sensor fusion up to 200 Hz. The Baiobit sensor has inter-instrument correlation coefficient ranging between 0.90 and 0.99, and an intra-instrument coefficient of variation of ≤2.5%, making it suitable for the assessment of physical activity with the same technical specifications of G-WALK (BTS Bioengineering, Garbagnate, Italy) [16,17,18].Two three-dimensional AMTI force platforms (dimensions: 464 × 508 × 82.5 mm; AMTI, Wetertown, MA, USA, sampling rate = 200 Hz).An optoelectronic system composed by an eight-infrared camera (BTS Bioengineering, Garbagnate, Italy).

### 2.3. Procedures

Thirty-eight passive reflective markers were placed on a selected anatomical landmark, according to the clustered base marker protocol, as shown in Figure 1. Anthropometric measurements included height, weight, anterior superior iliac spine (ASIS) width, leg length, knee width, ankle width, and pelvis depth. The Baiobit wearable sensor was fixed on the waist of the subject at the first and second sacral vertebra levels (S1 and S2) by means of an adhesive plate (Figure 2c).

All participants performed a 5-minute warm-up session in which the following exercises were executed in sequence: (i) 10 squats with hands on the hips × 2 repetitions; (ii) 6 forward lunges (3 on each side) × 2 repetitions; and (iii) low skip for 7 m × 2 repetitions. A 5 m distance was walked between each repetition.

Each participant was verbally instructed on the type of jump to be performed. In order to allow the subject to become familiar with the test and the marker set, two jump trials were performed before starting the recordings.

The protocol included the execution of 5 double-leg CMJs and 3 single-leg CMJs per side with hands on hips (the Akimbo technique) to minimize arm swinging. Each athlete performed 11 jumps and the trials order was randomized. One minute of rest was allowed between one type of jump and the next.

Before the acquisition the athletes’ first and last name, weight and height were entered into the software of the sensor (baiobit v1.2.6).

During the tests, the variables in Table 2 [19] were measured with the wearable sensor and stored by the software.

All data were collected using a frequency of 100 Hz and transmitted via a Bluetooth to a personal laptop (Notebook HP EliteBook 840 G3).

Simultaneously, the ground reaction forces and the marker trajectories were registered, respectively, by two three-dimensional AMTI force platforms and an optoelectronic system using an eight-infrared camera with a sampling rate of 100 Hz.

The clinicians involved in the study assessed the technique of jump execution and the subject was asked to repeat the test if it was not considered correct.

During the two-leg CMJs, each foot was placed on a force platform at a distance equal to shoulder width, whereas during the single-leg CMJs, the supporting limb was positioned within one of the two platforms (Figure 2a,b). All subjects wore standard running shoes when performing the jumps.

After the acquisitions, the subjects were asked to complete two questionnaires. The first called the international physical activity questionnaire (IPAQ), the Italian version, which assesses the daily physical activity, while the second consisted of specific questions in relation to the sport played (Table 1), the number of hours of training, and the sport level (category). The IPAQ (the Italian version) shows acceptable reliability properties and a satisfying–good consistency in the Italian adult setting [23], and the following procedure was used to interpret the IPAQ questionnaire, and the total metabolic equivalent (MET), given by the sum of MET from intense activity, the MET from moderate activities, and the MET from walking, was calculated. If the total MET was <700, then the subject was considered inactive. If the MET was between 701 and 2519, the subject was labelled as sufficiently active; if the MET > 2520, the subject was classified as very active. The MET from intense activities was given as 8 × minutes × days; the MET from moderate activities was given as 4 × minutes × days; and the MET from walking was given by minutes × days × 3 if the pace was moderate, × 3.3 if the pace was intense, and × 2.5 if the pace was slow [24].

### 2.4. Data Analysis

The parameters in Table 2 were manually imported from the Baiobit software into an Excel worksheet and the same parameters were computed through an ad hoc script in Matlab^®^ (R2020a MathWorks, Inc., Natick, MA, USA) and then plotted (Figure 3 and Figure 4) [25].

The jump height was calculated both by considering data from the force platforms alone and by using in combination data obtained from the optoelectronic system and the force platforms.

From the force platform data, the flight time was calculated and the jump height was obtained through the formula given in Table 2. Instead, using the vertical component of the three-dimensional coordinates of the PSIS marker and the events (take-off and landing) defined by the force platforms, it was possible to calculate the jump height with a simple sum.

### 2.5. Statistica Analysis

The normality of the data was checked through the Shapiro–Wilk test. Since the data have a normal distribution, they are reported as mean and standard deviation.

A two-tailed paired *t*-test was performed to assess the difference between the parameters computed in the Matlab^®^ script and in the sensor’s software. After that, a Pearson correlation coefficient was computed; however, by itself, this is not a good measure for determining agreement [26]. In fact, the Winer’s adjusted intra-class correlation coefficient (ICC; 2.1) with an absolute agreement and a 95% confidence interval (CI) was used to define the level of agreement between the gold standard (optoelectronic system and force platforms) and the inertial sensor system (Baiobit). The data demonstrate excellent reliability if the ICC is >0.90, good reliability if the ICC is between 0.75 and 0.90, moderate reliability if the ICC is between 0.75 and 0.5, and finally poor reliability if the ICC is <0.5.

The Bland–Altman plots were represented for all the considered variables (Figure 3 and Figure 4). The difference between the parameters computed in the Matlab^®^ script and in the Baiobit sensor’s software was reported on the y-axes, and the average between the gold standard and the Baiobit was reported on the x-axis. The systematic bias was the mean offset from zero, whereas the random error was defined as the difference between the upper and the lower limit of agreement (LoA) divided by two.

All the statistical analyses were performed in SPSS (IBM Statistic 26) and the level of significance was set at α = 0.05.

## 3. Results

Results are reported in Table 3 and Table 4. Comparing the two methods of measurements (force platforms vs. Baiobit—Table 3), the vertical mean heights for double- and single-leg CMJs with the Baiobit were 27.4 ± 5.7 cm (double-leg CMJs) and 12.6 ± 3.1 cm (single-leg CMJs), whereas they were 29.1 ± 6.3 cm (double-leg CMJs) and 12.0 ± 3.3 cm (single-leg CMJs) for the gold standard. Therefore, this resulted in a systematic bias of 1.8 cm for the two-leg CMJs and −0.6 cm for the single-leg CMJs. However, from the two-tailed paired t-test, no statistical difference was found.

Moreover, the CMJ flight heights of the Baiobit and the force platform were positively correlated; indeed, a high Pearson’s correlation coefficient was obtained (r = 0.92, *p* < 0.001 two-leg CMJs; r = 0.89, *p* < 0.001 single-leg CMJs) (Figure 3).

Bland–Altman plots of the two-leg and single-leg CMJ heights are shown in Figure 3. The values of the limits of agreement are −2.9 + 6.5 (CMJ bi) and −3.6 + 2.5 (CMJ mono), respectively.

In addition, the ICCs were classified as “excellent” for the double-leg CMJ heights (ICC = 0.92, 95% CI = 0.89–0.94) and “good” for the single-leg CMJ heights (ICC = 0.89, 95% CI = 0.85–0.91) (Table 3).

In contrast, the vertical mean double- and single-leg CMJ heights for the optoelectronic system and force platforms in combination are 26.5 ± 6.1 cm (double-leg CMJs) and 11.6 ± 3.3 cm (single-leg CMJs). Therefore, comparing this second method (optoelectronic system + force platforms) with respect to the Baiobit sensor results in a systematic bias of −0.9 cm for the two-leg CMJs and −1.2 cm for the single-leg CMJs. However, from the two-tailed paired *t*-test, no statistical difference was found.

Moreover, the CMJ flights heights of the Baiobit and the optoelectronic system and force platforms were positively correlated; indeed, a high Pearson’s correlation coefficient was obtained (r = 0.80, *p* < 0.001 two-leg CMJs; r = 0.83, *p* < 0.001 single-leg CMJs) (Figure 4).

Bland–Altman plots of two-leg and single-leg CMJ heights are shown in Figure 4. The values of the LoAs are: −8.2 + 6.4 (CMJ bi) and −4.6 + 2.3 (CMJ mono), respectively.

In addition, the ICCs were classified as “good” for both the double-leg CMJ heights (ICC = 0.80, 95% CI = 0.73–0.85) and for the single-leg CMJ heights (ICC = 0.86, 95% CI = 0.80–0.89) (Table 4).

Regarding the results obtained through the IPAQ questionnaire, in accordance with the calculation methods for METs, no athlete was inactive, as 9 out of 27 were sufficiently active, while the rest were very active (Table 1).

## 4. Discussion

The current study firstly aimed to validate the Baiobit wearable system against both the use of the AMNTI force platforms and the optoelectronic system in combination with force platforms, which were considered as reference methods in determining single-leg two-leg CMJ heights. Secondly, the study aimed to define the agreement between the previously mentioned measurement methods (force platforms vs. Baiobit; optoelectronic system + force platforms vs. Baiobit).

Regarding the first comparison (force platform vs. Baiobit), we found “excellent” reliability (ICC = 0.92) for the two-leg CMJs and a “good” reliability for the single-leg CMJs (ICC = 0.89), as well as a positive bias (1.8 cm) for the two-leg CMJ heights and a negative bias (−0.6 cm) for the single-leg CMJs. Hence, the Baiobit sensor underestimates the two-leg CMJs and overestimates the single-leg flight height. Only 6 of 135 jumps fell outside the LoAs for bipodal CMJs, while 8 of 162 were found for monopodal CMJs (Figure 3).

By considering the second comparison (optoelectronic system + force platform vs. Baiobit) instead, we found “good” reliability for both the two-leg CMJs (ICC = 0.80) and the single-leg CMsJ (ICC = 0.86), as well as a negative bias for the two-leg CMJ heights (−0.9 cm) and the single-leg CMJ heights (−1.2 cm). Hence, the Baiobit sensor overestimates the two-leg and the single-leg CMJ flight heights. Only 6 of 135 jumps fall outside the LoAs for bipodal CMJs, while 9 of 162 were found for monopodal CMJs (Figure 4).

Our findings are in accordance with many studies found in the literature investigating the validity of inertial measurements systems for determining two-leg CMJ heights [11,12,13,14,27]. McHugh et al. [27] observed that two-leg CMJ flight heights were strongly correlated between the optoelectronic system and an inertial sensor, but the latter underestimates the flight height values (1.8 ± 1.8 cm). Picerno et al. [13] found a low bias (0.6 cm) comparing a 3D inertial measurement unit with stereophotogrammetry. A study performed by Gökhan Yazici et al. [14] shows the reliability of a wearable movement analysis system (G-walk) on gait and jump assessment in healthy adults; all the jump parameters had excellent test–retest reliability (ICC: 0.90–0.97). Edward R. Brooks et al. [28] also evaluated the reliability and validity of CMJ height measurements using inertial measurement units compared with force platforms (correlation r = 0.95, reliability ICC = 0.91). Contrarily, the only system did not find good agreement in the Gyko inertial sensor system that alone was not considered an interchangeable device with force platforms [3].

To the best of our knowledge, no studies evaluated the reliability and the agreement of single-leg vertical CMJs, although it is currently being re-evaluated as an exercise to evaluate return to sport for the athlete [2]. We therefore think that more attention should be paid to single-leg CMJs. With regard to the Bland–Altman plot, there is no LoA value for which the measurement can be defined as agreeing or disagreeing with the gold standard [26]. Interpreting the results depends on the reference context; therefore, one must ask the athletic trainer or coach whether an underestimate of −1.2 cm or an overestimate of 1.8 cm is considered tolerable in an athlete’s training [29]. If the bias values are deemed acceptable, Baiobit can be considered a tool that can replace the use of force platforms and optoelectronic systems.

Regarding the IPAQ, 66% of the athletes were classified as very active (Table 1) and, on the questionnaire of specific questions in relation to the sport practiced (Table 1), more than 70% of the athletes practiced sports with similar jumping skills, such as volleyball, beach volleyball, and basketball [30]. To validate the sensor, it is necessary that the technique of performing the jumps is correct (e.g., avoiding lower limb flexion before landing). Administering the IPAQ could be a way to assess activity levels and then select athletes who are able to perform with the proper technique on the test.

### 4.1. Limitations and Possible Bias

The main limitation of the study is the heterogeneity of the population, as we analyzed athletes who practiced different sports and the sample contained a small number of female athletes (7 F vs. 20 M). Furthermore, the use of motion equation for the determination of jump height may be a possible source of error when comparing the agreement between the methods (force platforms vs. Baiobit; optoelectronic system + force platforms vs. Baiobit—Table 2).

Moreover, the system was attached by means of a plate covered with adhesive material on one side and a system of two snaps on the other, to which the device was anchored. One possible source of error may be due to detecting accelerating soft-tissue movements with the Baiobit sensor, as opposed to with force platforms [31]. It is extremely difficult, if not impossible, to remove all these motion artefacts.

### 4.2. Future Developments

Another possible evolution of the study could be to focus on a single sport to reduce variability in the activity and increase the size of the sample to be tested. In addition, there is a need to validate all the other types of tests proposed by the software, such as the walk test, the cervical, the trunk test, the shoulder test, the balance test (static, falling index), the squat jump, the drop jump, and finally the timed-up-and-go test.

## 5. Conclusions

The present study evaluates the agreement between measurements performed using the Baiobit inertial sensor system and force platforms alone or in combination with the optoelectronic system during single-leg and two-leg CMJs in athletes. The Baiobit sensor was found to underestimate the two-leg CMJs compared with force platforms alone and overestimate two-leg CMJ heights compared to the combination of the optoelectronic system and force platforms. Finally, the single-leg flight height, compared to both methods, force platforms, and optoelectronic system + force platforms, was overestimated. Thus, from a clinical point of view, the system is advantageous as it is cheaper and more compact than an optoelectronic system and/or force platforms, is transportable, and is therefore recommended in the world of rehabilitation and sports analysis.

## Figures and Tables

**Figure 1 ijerph-19-14720-f001:**
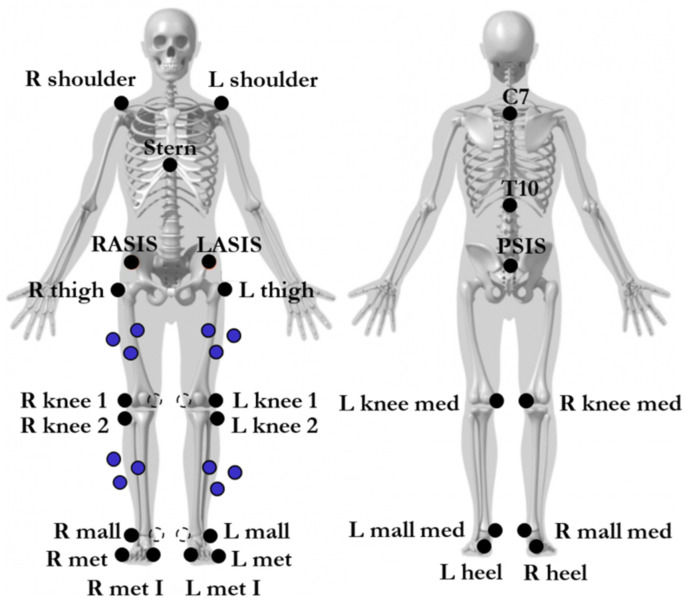
Cluster-based marker protocol with a 38 marker set. R shoulder, on the right acromion; L shoulder, on the left acromion; Stern, xiphoid process of the sternum; RASIS, on the right anterior superior iliac spine; LASIS, on the left anterior superior iliac spine; R thigh, on the center of the right great trochanter; L thigh, on the center of the left greater trochanter; R knee 1, on the lateral condyle of the right femur; L knee 1, on the lateral condyle of the left femur; R knee med, on the medial condyle of the right femur; L knee med, on the medial condyle of the left femur; R knee 2, on the lateral epicondyle of the right tibia; L knee 2, on the lateral epicondyle of the left tibia; R mall, on the center of the right lateral malleolus; L mall, on the center of the left lateral malleolus; R mall med, on the center of the right medial malleolus; L mall med, on the center of the left medial malleolus; R met, on the fifth metatarsal of the right foot; L met, on the fifth metatarsal of the left foot; R met I, on the first metatarsal of the right foot; L met I, on the first metatarsal of the left foot; R heel, on the right heel; L heel, on the left heel; C7, on the spinous process of the seventh cervical vertebra; T10, on the tenth thoracic vertebra; PSIS, on the mid-point of the right and left posterior superior iliac spine.

**Figure 2 ijerph-19-14720-f002:**
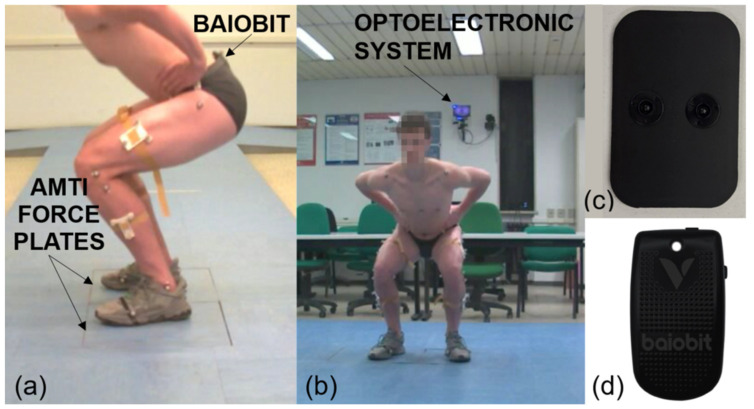
(**a**) Double-leg CMJ frame of the sagittal plane from the optoelectronic system with visible AMTI force plates; (**b**) double-leg CMJ frame of the frontal plane form the optoelectronic system with visible cameras; (**c**) bioadhesive plate; (**d**) Baiobit sensor (Rivelo Srl—BTS bioengineering Group, Milan, Italy).

**Figure 3 ijerph-19-14720-f003:**
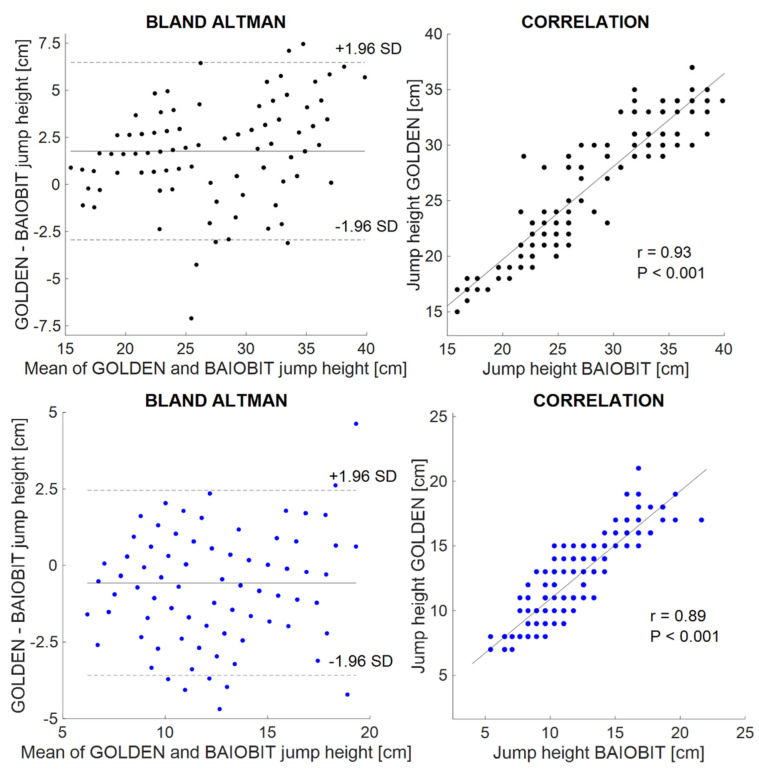
(**Top**) Double-leg CMJs on the left of the Bland–Altman plot comparing the force platform data (gold standard) vs. the Baiobit on the right of the correlation between the force plate and the Baiobit. (**Bottom**) Single-leg CMJs on the left of the Bland–Altman plot comparing the force platform data (gold standard) vs. the Baiobit on the right of the correlation between the force plate and the Baiobit.

**Figure 4 ijerph-19-14720-f004:**
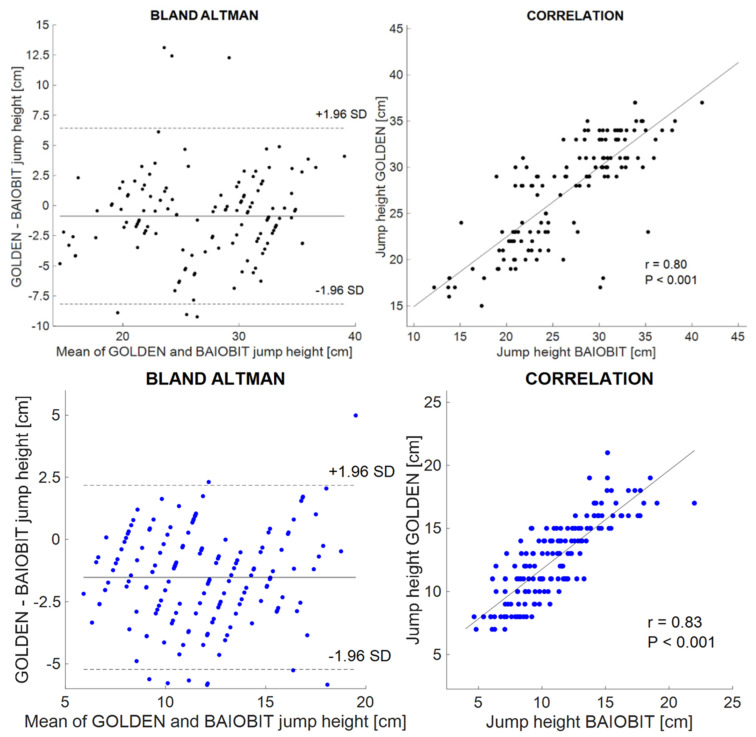
(**Top**) Double-leg CMJs on the left of the Bland–Altman plot comparing the optoelectronic system + force platform data (gold standard) vs. the Baiobit on the right of the correlation between the optoelectronic system + force plate and the Baiobit. (**Bottom**) Single-leg CMJs on the left of the Bland–Altman plot comparing the optoelectronic system + force platform data (gold standard) vs. the Baiobit on the right of the correlation between the optoelectronic system + force plate and the Baiobit.

**Table 1 ijerph-19-14720-t001:** Subject anthropometric characteristics, sports performed, and IPAQ results.

Subject (ID)	Sex	Age (Years)	Weight (kg)	Height (m)	Sport	IPAQ
1	F	36	73	1.85	beach volley	VA
2	M	26	77	1.82	triathlon	SA
3	M	27	86	1.91	volleyball	VA
4	M	22	81	1.92	basket	VA
5	M	27	85	1.92	basket	VA
6	M	23	80	1.91	swimming	SA
7	M	28	97	1.91	jiu jitsu	SA
8	M	28	65	1.78	triathlon	SA
9	F	24	47	1.64	athletics	VA
10	M	23	92	2.02	volleyball	SA
11	M	21	62	1.85	volleyball	VA
12	M	21	69	1.81	tennis	VA
13	M	20	83	1.94	basket	VA
14	M	37	93	1.95	beach volley	VA
15	M	24	71	1.76	tennis	VA
16	M	36	76	1.89	volleyball	VA
17	F	19	69	1.86	volleyball	VA
18	F	31	76	1.90	volleyball	SA
19	M	41	80	1.83	triathlon	VA
20	M	38	75	1.83	beach volley	VA
21	F	22	59	1.74	volleyball	SA
22	F	19	68	1.82	volleyball	VA
23	M	24	87	1.97	volleyball	SA
24	M	39	72	1.82	beach volley	VA
25	M	21	70	1.86	volleyball	VA
26	F	21	54	1.68	volleyball	SA
27	M	23	85	1.92	basket	VA
MEAN ± SD	20 M—7 F	27 ± 7	75 ± 12	186 ± 0.09		18 VA—9 SA

VA: very active, SA: sufficiently active. SD: standard deviation.

**Table 2 ijerph-19-14720-t002:** Jump variables from Baiobit user manual.

Parameter	Definition	References
Flight time (s)	Time between take-off and landing	[20]
Contact time (s)	Time elapsing between the start of the jump and the take-off	[20]
Jump height (m)	1.266 × flight time^2^	[20,21]
Reactivity index (-)	Flight time/contact time	
Force take-off (KN)	Maximum value of the force during take-off Expressed in % of body mass	[22]
Impact ratio (-)	Force landing/force take-off	[22]

**Table 3 ijerph-19-14720-t003:** Validity of the AMTI force plates and Baiobit sensor for jump height.

	Double-Leg CMJ	Single-Leg CMJ
AMTI force plates [cm]	29.1 ± 6.3	12.0 ± 3.3
Baiobit sensor [cm]	27.4 ± 5.7	12.6 ± 3.1
Systematic bias [cm]	1.8	−0.6
Random error [cm]	±4.7	±3.0
Lower LoA	−2.9	−3.6
Upper LoA	6.5	2.5
ICC (95% CI)	0.92 (0.89–0.94)	0.89 (0.85–0.91)

^1^ *t*-test *p* < 0.05.

**Table 4 ijerph-19-14720-t004:** Validity of the optoelectronic system and AMTI force plates and Baiobit sensor for jump height.

	Double-Leg CMJ	Single-Leg CMJ
Optoelectronic system + AMTI force plates [cm]	26.5 ± 6.1	11.6 ± 3.3
Baiobit sensor [cm]	27.4 ± 5.7	12.6 ± 3.1
Systematic bias [cm]	−0.9	−1.2
Random error [cm]	±7.3	±3.5
Lower LoA	−8.2	−4.6
Upper LoA	6.4	2.3
ICC (95% CI)	0.80 (0.73–0.85)	0.86 (0.80–0.89)

^1^ *t*-test *p* < 0.05.

## Data Availability

For the availability of data and materials, please contact the corresponding author.

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
