# Peer review of "Validity of the Baiobit Inertial Measurements Unit for the Assessment of Vertical Double- and Single-Leg Countermovement Jumps in Athletes"

_ijerph, 2022, doi:10.3390/ijerph192214720_

Round 1

Reviewer 1 Report

Thank you for the opportunity to review the manuscript entitled "Validity of the Baiobit inertial measurements unit for the assessment of vertical double and single-leg countermovement jump in athletes". The question is of importance for practical application and therefore interesting for the readers. Inertial sensors are increasingly used for movement, posture and function diagnostics. Therefore, a validation against the gold standard of "3-D-optoelectronic systems" seems appropriate. 

Key points: The introduction adequately introduces the topic. The importance of jump diagnostics for clinical, athletic and therapeutic purposes is addressed. The methodology for conducting the study is well described and allows for replication. The procedures described and the methodological approach are state of the art. The statistical analysis is based on the correct procedures such as correlations, Bland-Altman plots, etc. Here, an extension with regard to AI procedures would have to be discussed. The presentation of results, including figures and tables, is logically structured and reflects the central results. The discussion addresses the pros and cons and reflects the results based on the literature. Overall, a very logical, structured and formally correct paper. One question is allowed. Wouldn't a sports science (e.g. Sports) or technical journal (e.g. Sensors) be more appropriate.

 Specific references: In section 2.2 Devices the measurement system "Baiobit: is comprised of a triaxial accelerometer with multiple sensitivity (±2, ±4, ±8, 110±16g), a triaxial magnetometer 13 bit (±1200 μT) and a triaxial gyroscope with multiple sensitivity levels (±250, ±500, ±1000, ±2000°/s)" should be described in more detail (information on measurement errors, deviations, drift, manufacturer etc.).

In line 202 it should be indicated which ICC was calculated (adjusted, unadjusted, normalized etc.).

Line 283 should be improved .... knowledge, , no studies...

Information on the measurement facts should be discussed further (tissue movements, drift in the gyroscope, bias, outliers in the Bland-Altman plot, deviations. 

Reviewer 2 Report

Dear Authors,

thank you for the submission of your manuscript in IJERPH. I have thoroughly read your study. It has been a great pleasure to review this study.

Article deals with an interesting and debated topic, it is well written, and an accurate and valid methodological setup has been used.

However, I have some observations on the paper.

General:

 Please, check the abbrevition

Abstract

L24 Please, specify the abbreviation “IPAQ”.

Key word

Please, add the vertical jump

Introduction

L83 Please, specify the abbreviation “ICC”

L86: Please replace srl with Srl

Please, add your hypothesis

Materials and Methods

L120 Please, specify the abbreviation ASIS

L167 Please, add the reference

Please, add how systematic bias and random error are calculated

Results:

L219 Please, explain “systematic bias”

In the table 3 and 4 explain “random error”

Discussion:

ok

Figures:

Figure 1: Please, in the caption explain where the markers were placed

Table

Table 3-4 Please add 1 t-test p < 0.05.
